# Orbitofrontal neurons signal sensory associations underlying model-based inference in a sensory preconditioning task

Brian F Sadacca[1], Heather M Wied[1], Nina Lopatina[1], Gurpreet K Saini[1], Daniel Nemirovsky[1], Geoffrey Schoenbaum[1,2,3]*

[1]Intramural Research program of the National Institute on Drug Abuse, NIH, Baltimore, United States; [2]Department of Anatomy and Neurobiology, University of Maryland School of Medicine, Baltimore, United States; [3]Department of Neuroscience, Johns Hopkins School of Medicine, Baltimore, United States

**Abstract** Using knowledge of the structure of the world to infer value is at the heart of model-based reasoning and relies on a circuit that includes the orbitofrontal cortex (OFC). Some accounts link this to the representation of biological significance or value by neurons in OFC, while other models focus on the representation of associative structure or cognitive maps. Here we tested between these accounts by recording OFC neurons in rats during an OFC-dependent sensory preconditioning task. We found that while OFC neurons were strongly driven by biological significance or reward predictions at the end of training, they also showed clear evidence of acquiring the incidental stimulus-stimulus pairings in the preconditioning phase, prior to reward training. These results support a role for OFC in representing associative structure, independent of value.

DOI: https://doi.org/10.7554/eLife.30373.001

*For correspondence:
geoffrey.schoenbaum@nih.gov

## Introduction

Using knowledge of the structure of the world to infer value is at the heart of model-based reasoning, and relies on a circuit that includes the orbitofrontal cortex (OFC) (*Stalnaker et al., 2015*; *Rudebeck and Murray, 2014*; *Wallis, 2011*). When OFC is intact, rats and primates can use the causal structure of their environment to infer the value of elements on-the-fly. With OFC inactivated or lesioned, they cannot. This is evident in a variety of situations (*Gallagher et al., 1999*; *Izquierdo et al., 2004*; *Reber et al., 2017*; *Gremel and Costa, 2013*; *West et al., 2011*; *Takahashi et al., 2009*; *McDannald et al., 2005*; *Walton et al., 2010*), however it is perhaps most striking during sensory preconditioning. Here, inactivation of the OFC entirely and selectively impairs the use of previously acquired stimulus-stimulus associations to guide responding when one of the cues later comes to predict food (*Jones et al., 2012*).

How might the OFC support such inference? Some proposals focus on the ability of OFC neurons to respond to cues based on their acquired biological significance or value (*Padoa-Schioppa and Assad, 2006*; *Padoa-Schioppa, 2011*; *Rolls, 1996*; *Levy and Glimcher, 2012*; *Rolls et al., 1996*; *Rolls and Grabenhorst, 2008*; *Kringelbach, 2005*). The loss of such signaling is proposed to affect value-guided behavior. However, inactivation or lesions of OFC typically only affect value-guided behavior that requires inference or model-based processing (*Schoenbaum et al., 2011*). If the value can be derived from direct experience, the OFC is not normally necessary. This raises the possibility that the OFC is required for representing the model and perhaps not, uniquely, for encoding value

(*Wilson et al., 2014*; *Schuck et al., 2016*). A clear distinction between these two accounts comes when there are associations to be learned among neutral or valueless cues. If the core function of the OFC is to represent associative information that has biological significance or value, then this area should not represent such neutral associations until they have acquired some significance. On the other hand, if the core function of the OFC is to represent the causal structure of the world, then one might expect to see these relationships represented in some manner, even before they have any significance.

Here we directly tested these predictions by recording OFC neurons in rats during sensory preconditioning (*Brogden, 1939*). In this task, hungry rats are initially exposed to pairs of neutral cues (A->B, C->D). In subsequent conditioning sessions, the second cue in each pair is presented, one of which predicts a food reward (B->US, D). Finally responding to the first cue in each pair is assessed in an unrewarded probe test (A, C). As noted above, inactivation of the OFC in the probe test abolishes the normal increase in responding to A without affecting responding to B (*Jones et al., 2012*). If this is because of a role for the OFC in representing value, either independent of or combined with associative structure, then neural activity will reflect the significance of A and its relationship to subsequent events only in the probe test. By contrast, if this is because of a role for OFC in representing associative structure, independent of value, then neural activity in the OFC should reflect the relationship of A (and C) to subsequent events in both the probe test and the initial preconditioning phase.

## Results

We trained 21 rats with recording electrodes implanted in the OFC in a sensory-preconditioning task similar to the one used in our prior study (*Jones et al., 2012*). In the initial phase, rats learned to associate two pairs of 10 s auditory cues (A->B; C->D) in the absence of reward. As there was no reward, rats showed no significant responding at the food cup and no differences among the different cues (one-way ANOVA, $F_{(3, 80)}=0.54$, $p=0.66$; *Figure 1A*). In the second phase, rats learned that one of the auditory cues (B) predicted reward and the other (D) did not. Learning during conditioning was reflected in an increase in responding at the food cup during presentation of B, but not D (two-way ANOVA, main effect of cue: $F_{(1, 246)}=46.95$, $p<0.001$, main effect of session: $F_{(5, 246)}=11.75$, $p<0.001$ interaction: $F_{(5, 246)}=3.49$ $p=0.0046$; *Figure 1B*). In the final phase of the task, the rats were again presented with the four auditory cues, beginning with reminder trials of cue B and D followed by unrewarded presentations of cues A and C. As expected, the rats responded at the food cup significantly more to cue B than D (*Figure 1C*, left panel; t-test$_{BD}$: $t(20) = 8.23$) and more during presentation of A, the cue that predicted B, than during presentation of C, the cue that predicted D (*Figure 1C*, central panel; ANOVA, main effect of cue: $F_{(1, 251)}=5.79$, $df = 1$, $p=0.017$; t-test$_{AC}$: $t(20) = 2.15$, $df = 1$, $p=0.044$).

### Orbitofrontal neurons acquire ability to distinguish cue pairs during preconditioning

We recorded 266 neurons from OFC during the two preconditioning days (an average of 6 neurons per subject per day). Of these, 42% (112/266) significantly increased firing to at least one of the cues during preconditioning (right-tailed rank-sum between baseline and cue response, $p<0.05$), while 15% significantly decreased firing (40/266; left-tailed rank-sum, $p<0.05$). Overall, the prevalence of modulated firing to each of the individual cues was roughly equivalent (excited: 20% A, 18% B, 20% C, 13% D; inhibited: 7% A, 7% B, 4% C, 2% D).

This population included some neurons responding to one or both cue pairs, and such correlates were over-represented in the population of neurons responding to at least one of the cues, with elevated firing to both cues of a pair (A and B or C and D, 45/112) more common than elevated firing to cues of different pairs (A and D or B and C, 23/112; chi-squared test for independence, $X^2 = 10.2$; $p=0.0014$). This pattern is evident in *Figure 2A*, which plots the average (AUC) normalized responding of each of the 266 neurons to each preconditioned pair, ordered by how distinctly neurons responded to the initial cue in each preconditioned pair. This plot shows that those neurons that respond to one cue of a pair (e.g., cue A) have a strong tendency to respond to the other cue of a pair (e.g. B), confirming the pattern seen in individual neurons (*Figure 2B*). If this pattern was merely the result of neurons having a general sensitivity to auditory cues, we would expect the neurons that

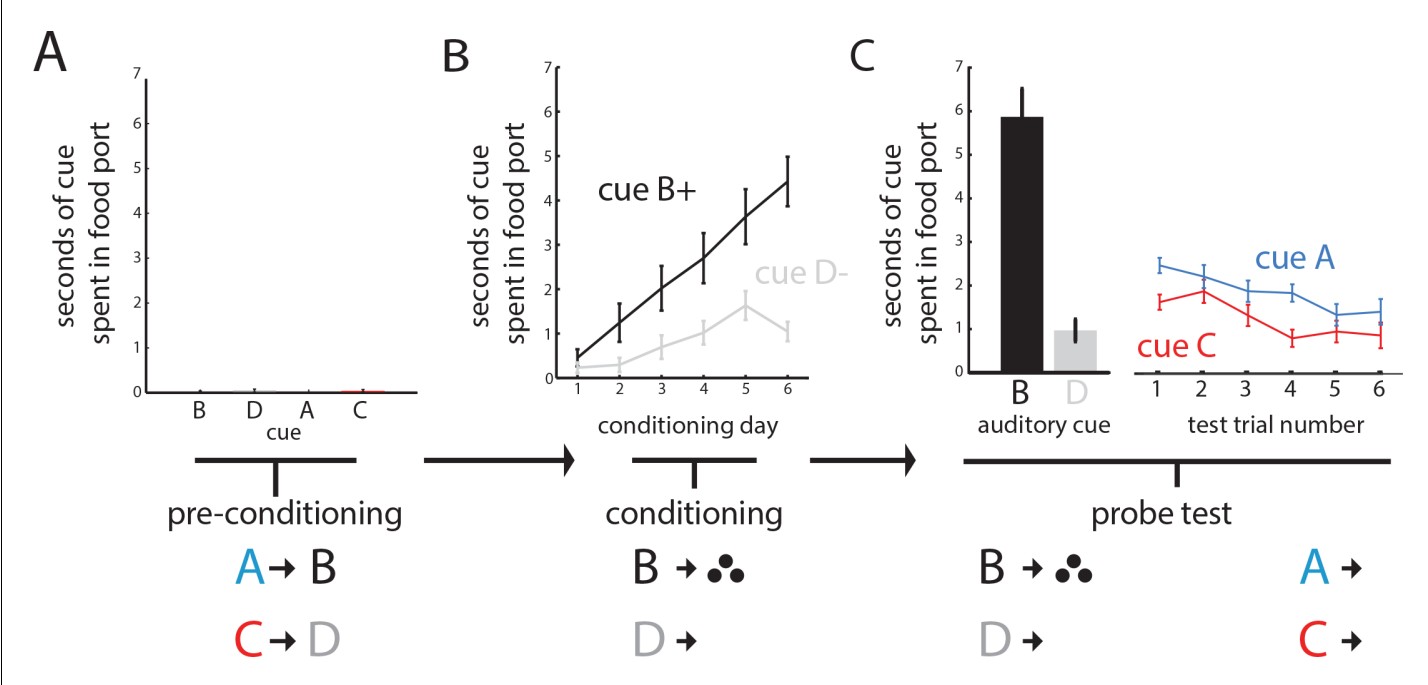

**Figure 1.** Rats learn to infer the value of a never-before rewarded cue in sensory preconditioning. Panels illustrate the task design and show the percentage of time spent in the food cup during presentation of the cues for each of the three phases of the sensory preconditioning task. (**A**) In an initial preconditioning phase, rats (n = 21) learned to associate auditory cues in the absence of reinforcement; during this phase there is negligible food cup responding. (**B**) In a second conditioning phase, rats learn to associate cue B with reward; conditioned responding progressively increases across sessions (displayed as mean and SEM). (**C**) In a final test, rats were presented with a reminder of conditioning trials, followed by presentation of the two 'unconditioned' cues A and C alone. Responding to cue A over cue C is evident in the averaged responding across rats (right, displayed as mean and SEM; one way ANOVA across cues A and C, p>0.05).
DOI: https://doi.org/10.7554/eLife.30373.002

fired to one cue pair to also fire to the other cue pair. However, the strength of response to one cue pair (e.g., A and B) tended to not be strongly predictive of a response to the other cue pair (e.g., C and D). To test whether this pattern was statistically reliable, we examined the relationship between the mean spiking above baseline to each cue between the paired cues and between the cues that were not paired for all 266 neurons recorded in both days. As illustrated in *Figure 2C*, we found that OFC neurons were much more likely to have a similar response to paired cues (AB or CD) than to unpaired cues (CB, AD). This was true across all neurons (n = 266 $rho_{AB}$=0.74 and $rho_{CB}$ = 0.16, $Zr1-r2$ = 9.05, $p<10^{-16}$; $rho_{CD}$ = 0.75, $rho_{AD}$ = 0.23, $Zr1-r2$ = 8.59, $p<10^{-16}$). Thus, OFC neurons tended to respond similarly to the paired auditory cues and distinctly to each of the pairs.

We next tested if the correlated firing during the contiguous cues was merely the result of their temporal adjacency. If this is the cause, then nearby bins should be more correlated than temporally distant bins. The supplement to *Figure 2* tests this, comparing the mean correlation between activity in bins early (first half) and late (last half) in one cue of a pair to activity in the other cue of the pair. While there is an overall lower correlation (owing to more bin-to-bin variation in firing rates of individual neurons), the influence of timing on correlation is, at best, surprisingly modest, and formally there is no significant difference between the strength of these correlations calculated with the early versus the late bins for either set of cues on either day. These results suggest that mere temporal contiguity of the time bins does not account for the correlated firing observed in OFC during the cues in preconditioning.

To say that this correlation is a measure of the association of the cues, however, something about this correlation should grow or change across preconditioning. To assess this, we examined how these correlations evolved during learning in neurons from rats that demonstrated they learned the relevant sensory association by responding more to cue A than to cue C in the final probe test (n = 203 from 14/21 rats). The outcome of this analysis is displayed in *Figure 3A*. As expected, there

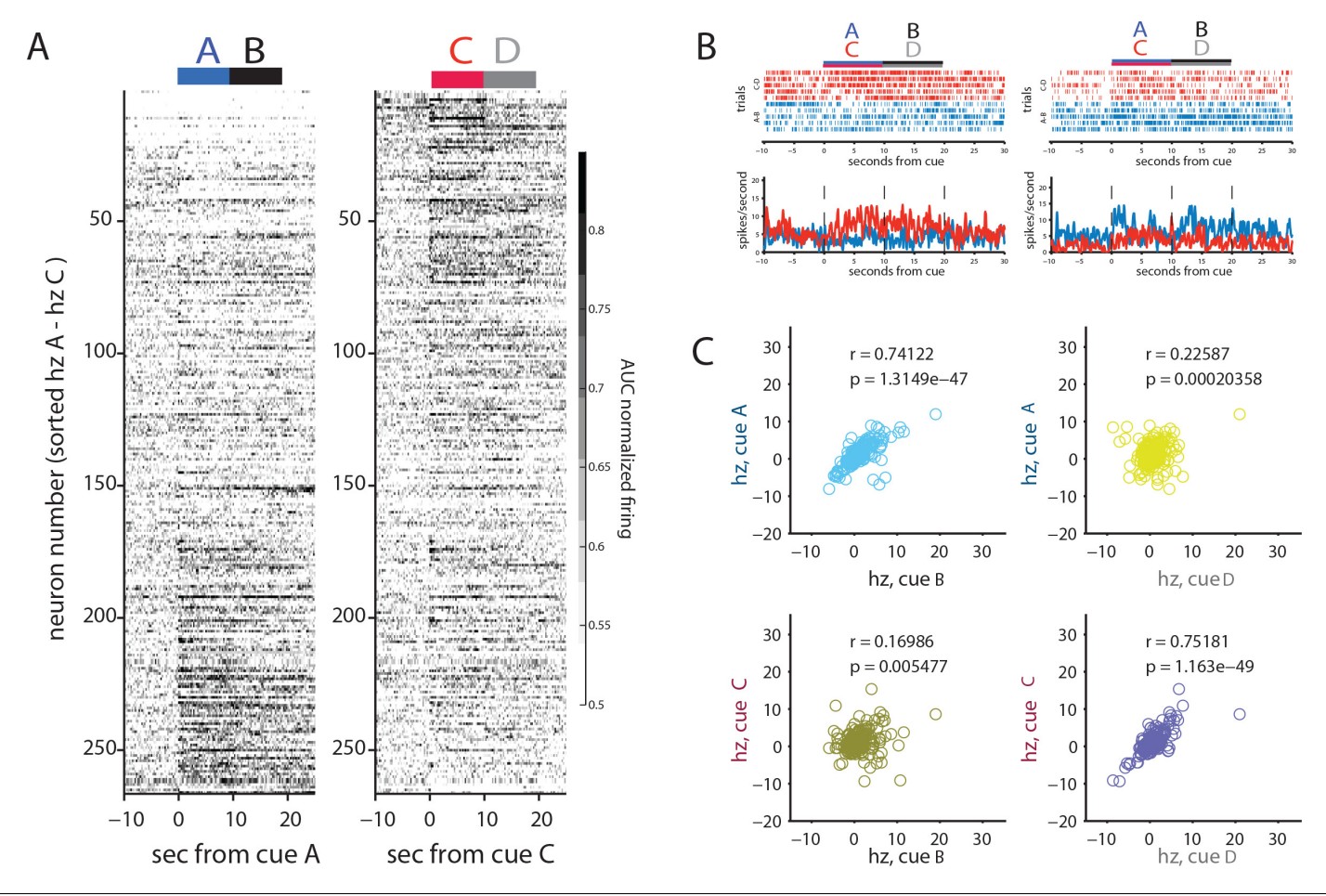

**Figure 2.** Orbitofrontal neurons encode preconditioned pairs in the absence of reward. (**A**) AUC normalized responding of all 266 neurons recorded across the two days of preconditioning for either A-B trials (blue, left) or C-D trials (red, right), sorted by for the relative response to cue pairs (cues AB vs CD). The plots show that different neurons seem to fire to the AB pair or the CD pair. (**B**) Cue-evoked firing in two individual neurons shows differential firing to either the AB or CD pair. (**C**) Correlations between individual neural responses to paired or unpaired cues above the neuron's average responding. Plots reveal much greater correlated firing between paired than unpaired cues during preconditioning (A-B, top left; C-D, bottom right).

DOI: https://doi.org/10.7554/eLife.30373.003

The following figure supplement is available for figure 2:

**Figure supplement 1.** The correlation between pairs of cues is not solely determined by temporal contiguity.

DOI: https://doi.org/10.7554/eLife.30373.004

was a strong positive relationship between firing to the paired cues (AB and CD), and no relationship between firing to the unpaired cues (AD and CB). Furthermore, the pattern of this correlation differed across days: on day 1, the correlations were strongest on the same trial for each cue of a pair, weaker for adjacent trials of that pair, and negligible between the early trials of one cue of the pair and the late trials of the other cue of the pair. This pattern of relatively restricted correlation is consistent with the contiguity explanation – correlations do not reflect a consistent representation of the pair but are merely caused by a subset of neurons that happen to be activated by adjacent sounds at a particular time. However on day 2, following a full day of preconditioning and time to consolidate associations, the correlations between cues of a pair encompass most of the 6 trials of the opposite pair of each cue, forming more of a checkerboard pattern, as if a reliable response is evoked to each cue of a pair. The across-trial reliability of the evoked response is consistent with identification of the cue pairs as a reliable feature of the environment in these rats.

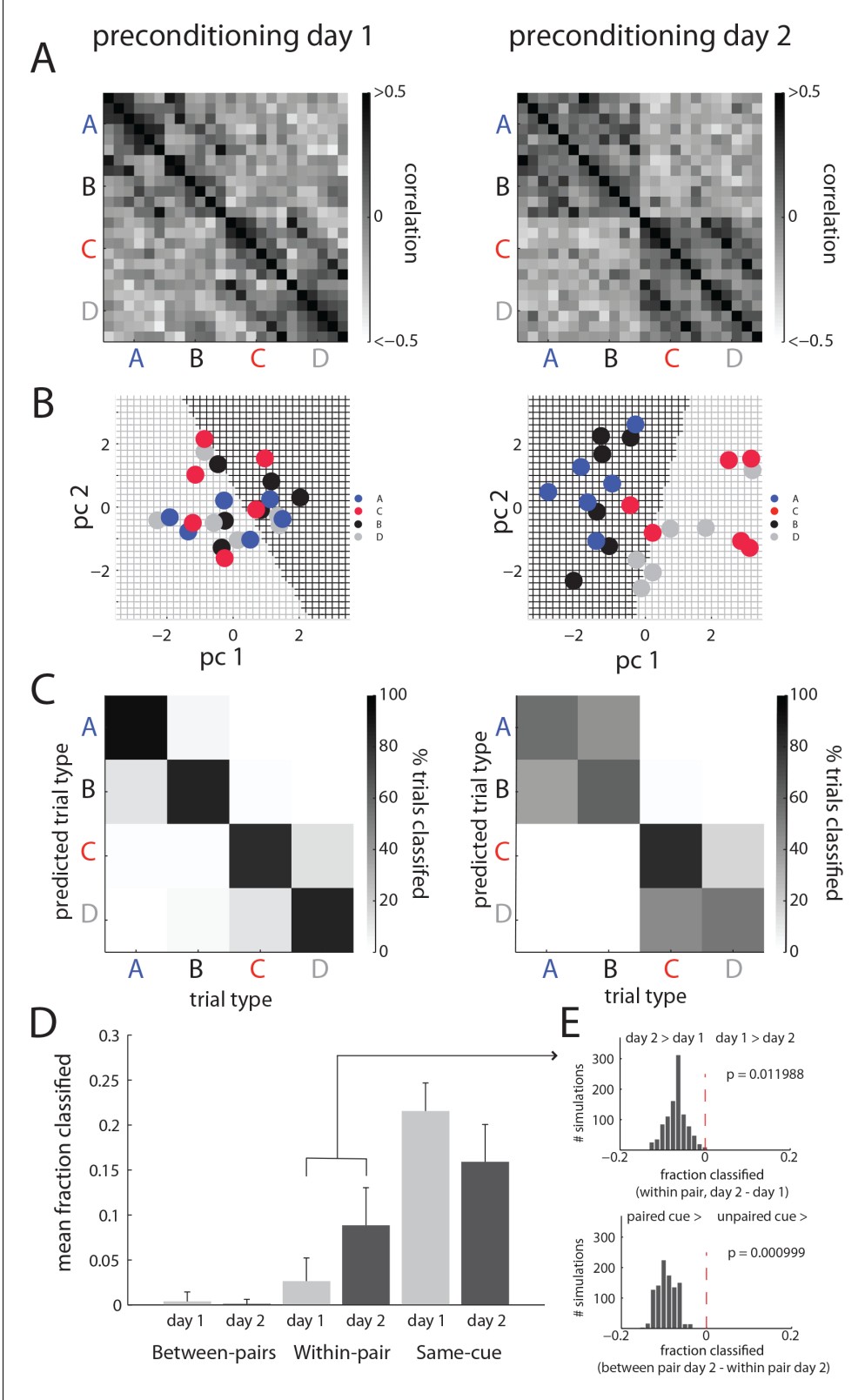

**Figure 3.** Orbitofrontal neurons ability to reflect neutral associations becomes more reliable across conditioning. (**A**) Pearson correlation of individual trials of OFC activity, calculated from all neurons recorded on preconditioning day 1 (left) or day 2 (right), shows that correlated firing between the paired cues spreads across trials conditioning (day 1 vs day 2). This spread does not occur for unpaired cues. (**B**) This effect is also evident in individual

*Figure 3 continued*

ensembles. An example of this is visualized for one ensemble of neurons in the two dimensions that best capture the population response from a principal components analysis on that ensemble from preconditioning day 1 (left) vs day 2 (right). On day 1, the ability to distinguish trial types via a linear discriminant classifier (indicated by the colored underlying grid; black indicating a likely B point, grey indicating D) does a much better job discriminating the paired cues (**A and C**) on day two than on day 1. (**C**) The classification illustrated in B is performed parametrically across randomly sampled pseudo-ensembles equal to the size of the population recorded on that day with replacement, and the classification of individual trials is displayed as a confusion matrix for all possible pairwise comparisons (e.g. cue A labeled as A, B, C or D). There is a notable decrease in correct classification and an increase in mis-classification within cue-pairs (e.g. cue A labeled as cue B) across days, resembling the results in panel A. (**D**) These results were then aggregated by error type (within or between pair) vs correctly labeled trials (mean ±SEM across 1000 resampled ensembles) to confirm the increase in within-pair classification across days. (**E**) Permutation tests performed on resampled ensembles showed that the increase in within-pair classification across days was unlikely to be obtained by chance.

DOI: https://doi.org/10.7554/eLife.30373.005

If OFC responses to paired, innocuous cues become more reliably similar, we should be able to identify OFC's response to one pair of cues on a given trial better on the second preconditioning day than on the first, when the correlation among trials is less consistent. For example, *Figure 3B* displays the relationship in firing within the neurons recorded in a single session for presentations of each cue, plotted as the first two principal components of the population response on each of the two preconditioning days. On day one the ability to classify trials as B (black grid background) or D (grey grid background) does not discriminate the paired cues (A and C) very well, whereas the ability to classify B and D on day two is nearly perfect at telling their paired partners apart.

To test this quantitatively, we generated pseudo-ensembles for each preconditioning day. We modeled the population response with a simple linear discriminant classifier trained on all but one response to each of the cues and then tested the ability of this model to classify the held-out presentation of each cue. The held-out trials (one each of A, B, C, and D) could then be labeled as having come from any one of the cues. To establish the reliability of this classification, this analysis was repeated on 6 sets of cue presentations, and on resampled ensembles (with replacement) of size equal to the population recorded that day from rats that learned the task (89 neurons for day 1 and 114 neurons for day 2) one thousand times. *Figure 3C* illustrates the average output of this classifier as a confusion matrix, with 'correct' classification (responses to a cue labeled as that cue) on the main diagonal, and different kinds of mis-classification along the other diagonals, with trials sometimes categorized as a 'within-pair' error (e.g., labeling an A trial as coming from cue B), or a 'between-pair' error (e.g., labeling an A trial as coming from cue C or D). While between pair errors were relatively rare, it appears that on average there is a substantial increase in within-pair errors from day 1 to day 2. When the output of these classifiers are aggregated by response (correct, or within and between pair errors), displayed in *Figure 3D*, the population response showed a decline in self-classification and an increase in within-pair classification across the two preconditioning days. This shift in the distribution of errors in classification is consistent with the expectation that if cues of a pair are being represented more similarly across trials, there should be an increase in within-pair misclassification. To test whether a shift this large could have occurred by chance, we performed a permutation test where the distribution of the shift in between-type errors from day 1 to 2 was computed across all resampled ensembles. According to this approach, which allows the direct calculation of a p-value for the specific difference that was observed, the shift in within-pair classification across days was unlikely to occur by chance (p=0.009, *Figure 3E*, top panel). A similar permutation test on the difference between the within pair and between pair classification on day two found that this difference was also unlikely to occur by chance (p=0.0001, *Figure 3D*, top right panel).

Finally to control for baseline differences between trials, as some neurons distinguish AB trial blocks from CD trial blocks, we repeated this classification analysis, either by simply by subtracting baseline firing on individual trials from the cue responses on that trial as a first control dataset or by fitting a regression model to the relationship between cue firing on a given trial and firing at baseline on that trial and using the residuals from that regression a second control dataset and classifying both control datasets as above. In both, we again observed an increase in within-pair classification

from day 1 to day 2 ($p_{subtraction}$ = 0.001; $p_{residual}$ = 0.007) and a greater within-pair than between pair classification on day 2 ($p_{subtraction}$ = 0.011; $p_{residual}$ = 0.038).

## Orbitofrontal neurons acquire the ability to predict reward during pavlovian conditioning

As noted earlier, one hallmark of OFC neurons is they acquire responses to cues that have biological significance or value through pairing with reward. Accordingly, we found that activity to B increased significantly in the 683 neurons recorded over the course of 6 days of conditioning. The evolution of this increase can be seen in the average (AUC) normalized responding of these neurons to cues B and D shown in *Figure 4A and B*. Firing to cues B and D is initially very similar, however over the 6 days of training, cue B comes to evoke a larger neural response than cue D. Although firing to B is contaminated by the delivery of reward at several points within the cue, the increased firing is also evident in many neurons at the outset of cue B. On the final conditioning day, twice as many neurons fired above baseline in the first 2 s of cue B, before reward onset, than did so at the outset of cue D (17%, 17/101 vs 7%, 7/101; $X^2$ = 4.73, p=0.03). In addition, the prevalence of such neurons increased significantly over the course of conditioning for rewarded cue B (17% or 17/101 on day 6 vs 8% or 10/128 on day 1; $X^2$ = 4.41, p=0.036) vs cue D (7% or 7/101 on day 6 vs 6% or 8/128 on day 1; $X^2$ = 0.04, p=0.84). This increase is similar to what we have observed previously in similar settings (*Takahashi et al., 2013*; *Lucantonio et al., 2014*).

## Orbitofrontal neurons exhibit ability to infer reward in the probe test

Given the increase in the fraction of neurons firing to B across conditioning, we wondered whether the pattern of neural activity to the other cues paired with them in preconditioning might also change. This would be consistent with a role for OFC in dynamically representing the current cognitive map (rather than some prior, static one). To examine this, we plotted the activity of the 205 neurons (averaging 9.8 neurons per subject) recorded in the probe session. Recall that during the probe test in the current experiment, we presented cues B and D in a reminder phase with reward given,

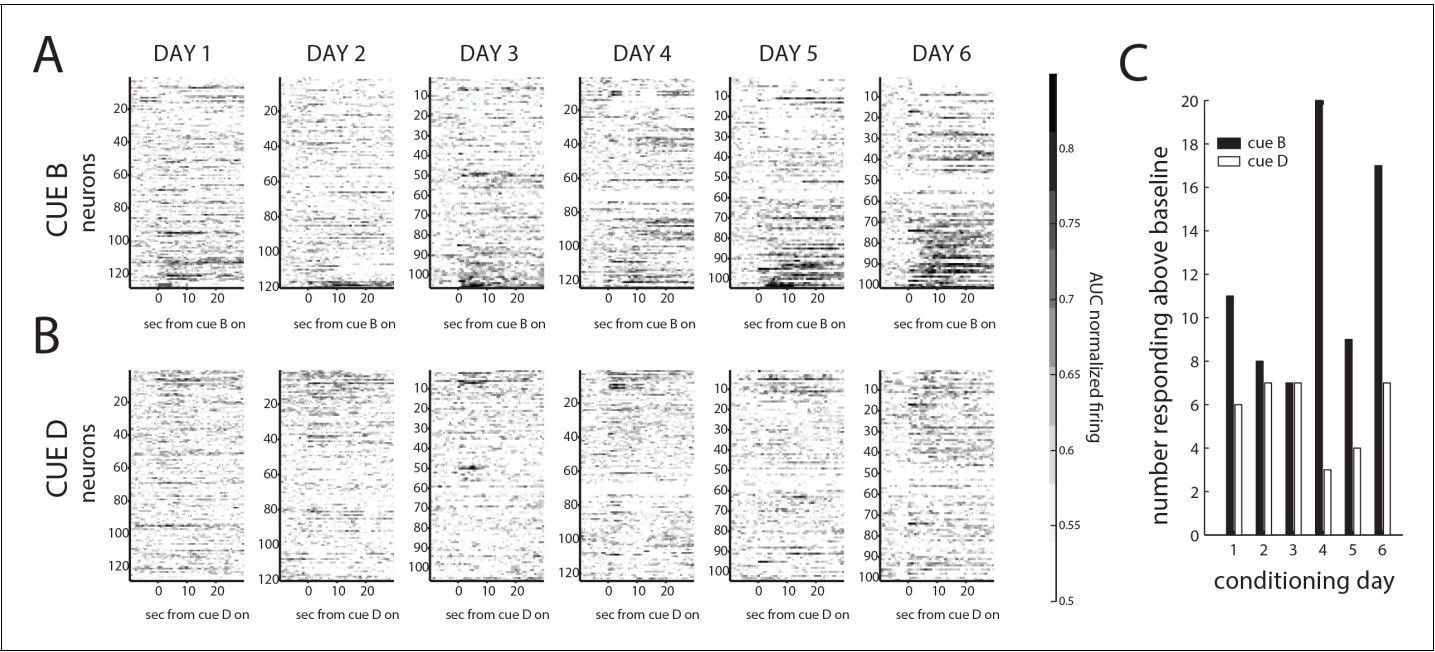

**Figure 4.** Orbitofrontal neurons accumulate responding during conditioning. (**A**) Normalized responding to cue B and reward (ordered by their relative responding to cue B vs cue D) shows an increased fraction and diversity of responses over the course of the six conditioning days, while (**B**) normalized responding to cue D on each conditioning day shows more modest changes across conditioning. (**C**) These differences are evident in the fraction of neurons responding to each cue across the 6 days of conditioning. There were significantly more neurons responding to cue B in the final day of conditioning than the first (p>0.05, chi-squared test), with no significant change in the fraction responding to cue D.
DOI: https://doi.org/10.7554/eLife.30373.006

and then followed this with unrewarded presentations of the paired cues, A and C. Consistent with the conditioning data, a larger fraction of neurons again exhibited increased activity to the rewarded cue B than cue D (31% vs 8%; one-way sign-test baseline vs. cue, *Figure 5A*). However, in addition, the fraction of neurons responding above baseline to the preconditioned cues (A and C) also increased significantly (*Figure 5A*). Notably, although the firing to each remained largely segregated, the increase was seen to both cues, with 37% of neurons elevating their firing rate to cue A and 35% of neurons elevating their firing rate to cue C (across first 3 trials of each for comparison with B/D fractions, one-way sign-test, baseline vs. cue, $p<0.05$), with roughly the same fraction inhibited as in preconditioning (6% for cue A and 7% for cue C). While some of this increase may reflect generalization, the reorganization favored the promotion of firing correlates that reflected the earlier learning. This is evident in *Figure 5B and C*, which plot the mean normalized response of the ten percent of neurons with the largest difference in responding to cue A over C (*Figure 5B*) or vice versa (*Figure 5C*). In neurons with the stronger response to A, there is a strong and prolonged response to cue B (and reward), whereas in neurons with the stronger response to C, there was only a modest response to cue B, and this response is primarily observed only after reward delivery begins. These distinctions hold for both more selective and permissive comparisons of A vs. C responding.

The increase in the fraction of neurons responding to cues A and C, which had not been presented since preconditioning, coupled with the preserved relationship between firing to cues A and B, shows that the activity of OFC neurons integrates associations formed in preconditioning and

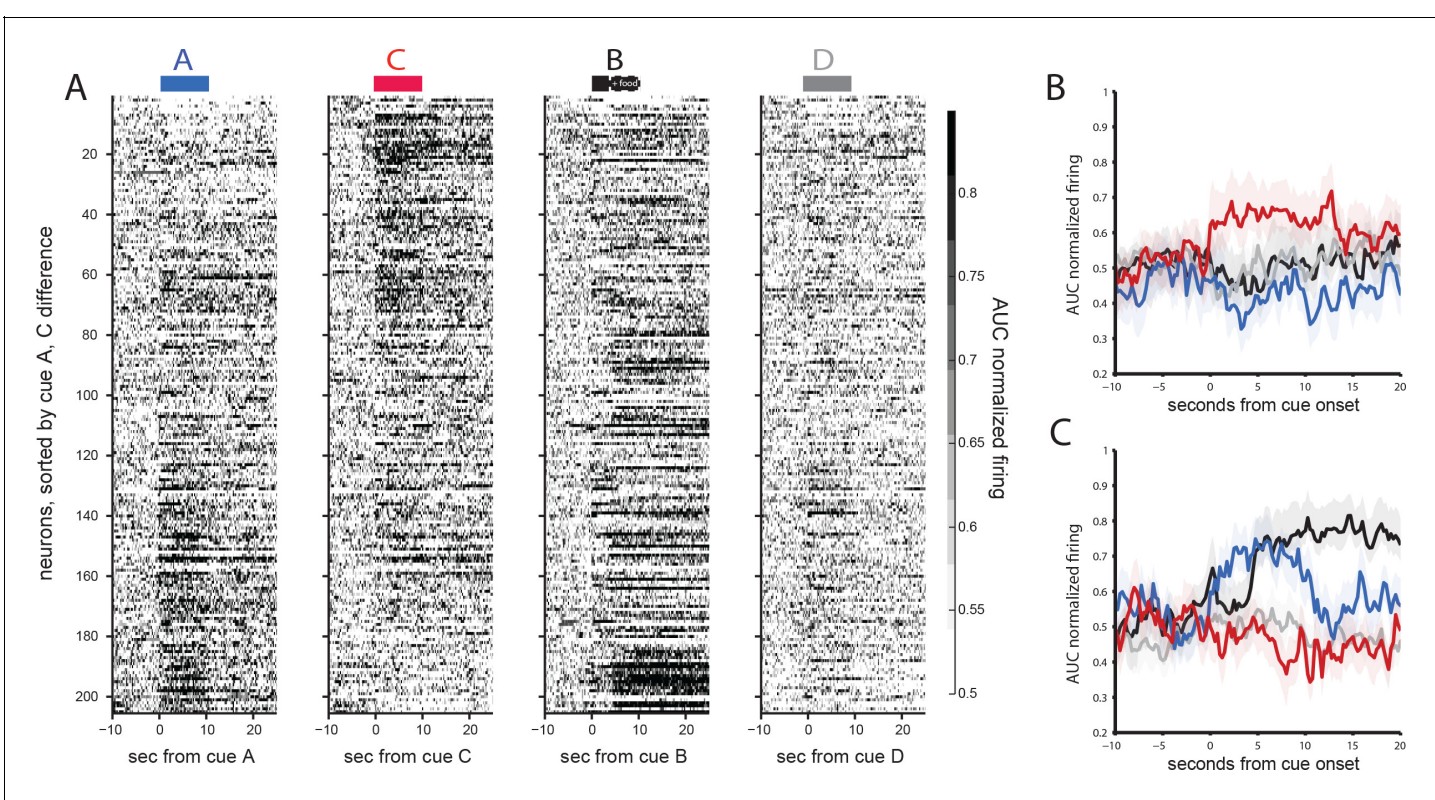

**Figure 5.** Orbitofrontal neurons distinctly encode preconditioned and conditioned cues in the final probe test. (**A**) Activity to cues A (blue), C (red), B (black), or D (grey), across all 205 orbitofrontal neurons during the probe test, sorted by their relative responding to cue A vs cue C. Plots show a distinct pattern of responding to cues A and C. In addition, the firing to cue B, now rewarded, is substantially higher than to any of the other cues. While the population response to cue B has changed substantially, there is still some similarity between responding to cue A and cue B, such that neurons that respond strongly to cue A are more likely to respond strongly to cue B than are neurons that respond strongly to cue C. This is made explicit when we isolate activity from the 10% of the neurons responding most strongly to one or the other cue. (**B**) Neurons responding most strongly to C have modest firing to cue B that is similar to the activity observed to the other cues. (**C**) By contrast, neurons responding most strongly to A have substantial and somewhat unique firing to cue B.

DOI: https://doi.org/10.7554/eLife.30373.007

conditioning in the probe test. As noted earlier, conditioned responding in this phase to cue A is OFC-dependent (*Jones et al., 2012*). To test whether the neural reorganization might be related to this dependence, we divided the recording data based on whether the rats showed evidence of pre-conditioning in the probe test. *Figure 6A* displays the relative activity between cues for the 150 neurons recorded in rats that responded more to cue A than to cue C. These neurons showed stronger correlated firing between formerly paired cues than between cues that had never been paired (n = 150, $rho_{AB}$ = 0.43 and $rho_{CB}$ = 0.19, $Z_{r1-r2}$= 2.27, p=0.023; $rho_{CD}$ = 0.37, $rho_{AD}$ = 0.12, $Z_{r1-r2}$ = 2.36, p=0.018). By contrast, *Figure 6B* displays the mean activity of 55 neurons recorded in rats that showed either no preference in responding to cues A and C or responded more to cue C than cue A. These neurons showed correlated firing between the unpaired cues that was as strong or stronger than that between the formerly paired cues (n = 55, $rho_{AB}$ = 0.45 and $rho_{CB}$ = 0.59, $Z_{r1-r2}$ = 0.90, p=0.36; $rho_{CD}$ = 0.12, $rho_{AD}$ = 0.14, $Z_{r1-r2}$ = 0.13, p=0.89).

To the confirm the robustness of the distinct patterns of correlations across trials and through time, we created another simple linear discriminant classifier, using pseudo-ensembles of 205 neurons, equal to the population recorded for that day, and trained using the mean activity evoked by the cues on A and C trials. We then asked this A/C classifier to identify activity during presentation of B or D to test whether firing to the preconditioned cues was, in essence, representing the subsequent cue in each pair. Because B had two phases, one before and one after the delivery of reward began, we conducted this analysis on segments of the trial, a 1 s window moved in 250 ms steps and iterated 1000x on resampled ensembles. The mean classification success was then compared to a null distribution created from the same classifier, with shuffled cue labels; classification better than 95% of the shuffled examples was labeled as significant (p>0.05). The result, plotted separately for the neurons recorded in good (*Figure 6C*) and poor (*Figure 6D*) performers, shows that above-chance classification (e.g. B = A and D = C) was only observed in ensembles composed of neurons from good performers. Further, the significant increase in correct classification came during the period when cue B overlapped with reward and was consistent through this period. This indicates not only that the ensembles reorganized in the good performers as a result of conditioning, but that they reorganized such that activity during A was best correlated with the middle and later sections of B, when reward could be expected to come. This is consistent with the idea that activity during A is directly signaling B and is association with reward, even though A was never presented with reward.

## Discussion

The OFC has long been implicated in our ability to respond adaptively and flexibly to obtain reward (*Gallagher et al., 1999*; *Izquierdo et al., 2004*; *Reber et al., 2017*; *Gremel and Costa, 2013*; *West et al., 2011*; *Takahashi et al., 2009*; *McDannald et al., 2005*; *Walton et al., 2010*; *Jones et al., 2012*). Traditionally this involvement has been linked to representing associative information of biological significance (*Rolls, 1996*; *Rolls et al., 1996*; *Rolls and Grabenhorst, 2008*; *Kringelbach, 2005*). More recently, research has emphasized the importance of the OFC to encoding the value or utility of available options, allowing decisions between them that reflect meaningful or idiosyncratic real-time changes in their desirability (*Padoa-Schioppa and Assad, 2006*; *Padoa-Schioppa, 2011*; *Levy and Glimcher, 2011*; *Plassmann et al., 2007*; *Padoa-Schioppa, 2009*; *Padoa-Schioppa, 2013*; *Tremblay and Schultz, 1999*; *Kobayashi et al., 2010*; *O'Neill and Schultz, 2010*). Together, these ideas have promoted the core function of the OFC as transforming information into an expectation of value (*Padoa-Schioppa, 2011*; *Levy and Glimcher, 2012*). However, an alternative view is that the OFC's core function is to represent a structure among environmental features, of which value is merely one of many features (*Stalnaker et al., 2015*; *Wilson et al., 2014*; *Schuck et al., 2016*; *Wikenheiser et al., 2017*). Here we tested between these different perspectives by examining the representation of associative information in OFC neurons and ensembles both before and after those associations had acquired biological significance. To do this, we recorded single unit activity in OFC during an OFC-dependent sensory preconditioning task (*Jones et al., 2012*). Activity was recorded during the initial preconditioning phase, while rats were exposed to neutral cue pairs, and subsequently during the probe test, when the same cues were presented after one had been paired with reward. As expected, we found that associative neural activity in the OFC was heavily driven by reward; the cue that had been paired with reward was

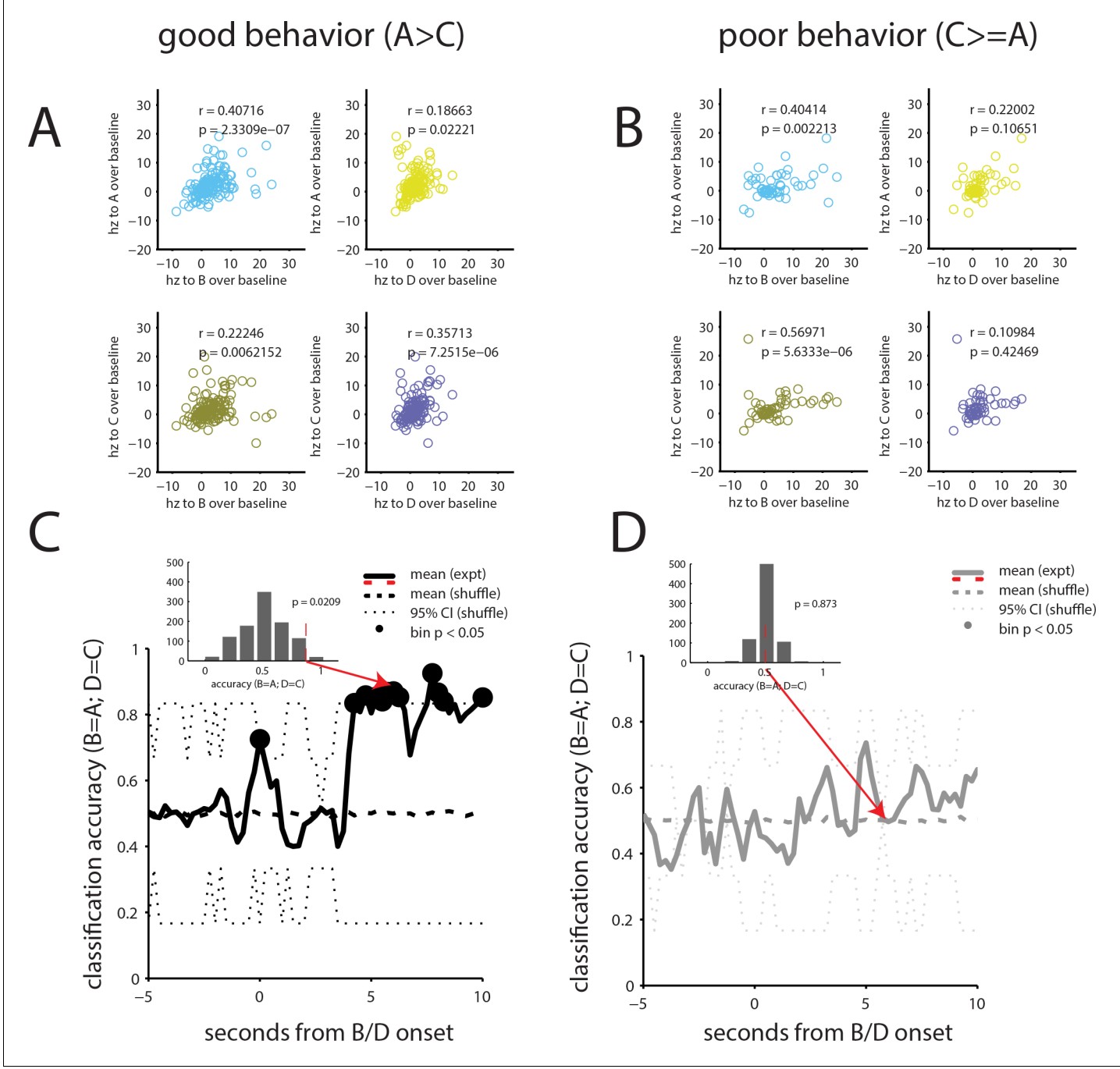

**Figure 6.** Orbitofrontal neurons signal preconditioned associations in probe test in rats able to infer expectations of value. (**A**) For the 150 neurons recorded in rats that showed evidence of preconditioning in the probe test, correlations between cues paired during preconditioning are well preserved and greater than between cues not paired during preconditioning (**B**) By contrast, for the 55 neurons recorded in rats that did not appear to precondition, the pattern is flipped, with greater correlations between the unpaired than the paired cues. (**C–D**) We attempted to classify trials based on this pattern of activity for rats that showed evidence of preconditioning (**C**) versus those that did not (**D**). For this, we trained a linear discriminant classifier on the evoked response of a pseudo ensemble of size equal to the population recorded (n = 205) to cues A and C and then tested the ability of this classifier to correctly identify the neural response to cues B and D. The mean success of this classifier at correctly identifying activity evoked by the paired cue was tested against that of a classifier trained and tested with shuffled cue labels (iterated 1000x, solid black line). The insets display the distribution of these results across iterations for one bin; classification in excess of 95% of shuffled resamples (dotted black line) was labeled significant (black circles). By this measure, classification accuracy for the ensemble recorded in rats that exhibited evidence of preconditioning was significantly above chance for the majority of bins during the second half of cue B, when cue B was co-presented with rewarding food pellets. By contrast classification accuracy for the ensemble recorded in rats that did not appear to precondition hovered near chance for all bins.
DOI: https://doi.org/10.7554/eLife.30373.008

strongly represented by the population. In addition, probe test firing to cues paired in preconditioning was strongly correlated, particularly in rats that showed evidence of preconditioning. However, while the OFC's response to these cues was robust once they were tied to an expectation of value, the response represented a modification of neural correlates of the arbitrary cue pairs evident and in fact acquired during the initial phase of training.

That OFC acquires neural representations of the arbitrary cue pairs in the initial phase of preconditioning, prior to the introduction of reward, suggests that the OFC builds associative representations even for information that does not have clear biological significance or value. While the implicit learning of statistical relationships between visual (*Turk-Browne et al., 2009*) or auditory cues (*McNealy et al., 2006*) has been reported in sensory cortices, it's striking that more frontal regions like OFC have access to these associations. In this regard, the OFC joins a growing number of associative regions, including hippocampal, retrosplenial, striatal, and even midbrain areas (*Cerri et al., 2014*; *Robinson et al., 2014*; *Sharpe et al., 2017a*; *Wimmer and Shohamy, 2012*), that appear to be involved in and even required for stimulus-stimulus learning.

But what is the actual role of these representations - if OFC is not simply signaling value, what does it signal? One possibility suggested by recent computational accounts is that correlates like these reflect a role in maintaining so called successor representations. These representations capture the expectation of moving to one state from another, independent of value, but stop short of encoding a full task model (*Gershman et al., 2012*). Successor representations have been applied to interpret neural activity in hippocampus (*Stachenfeld et al., 2017*), and aspects of these models would account for the apparent associative activity observed to the predictive cues (A and C) in preconditioning. While appealing, if OFC represents the matrix of future expected states, it is not clear why this activity changes as a result of conditioning to B. In simple versions of this model, an established matrix is not affected except by direct experience; A and C were not experienced again until the probe test, and yet the pattern of activity to cues A and C changed from preconditioning to probe. Alternatively, activity in OFC to A and C could reflect the product of their successor representation matrices and the value of the downstream states. This would explain the dramatic change in neural activity to A across conditioning, since the value of B was presumably altered by pairing with reward. However, responding to A does not seem to be fundamentally based on value cached in B, since that responding is affected by spontaneous changes in the value of the actual food (*Sharpe et al., 2017a*). Further, recent evidence shows that cue A in our design will not serve as a conditioned reinforcer, whereas a second-order cue will do so (*Sharpe et al., 2017b*). These data provide direct evidence that a preconditioned cue, at least in our design, is not accessing cached value by any common definition. While these disparate findings can perhaps be reconciled with successor representations models that incorporate off-line rehearsal or other additional processing steps, the activity we observe here seems more consistent with the proposal that the OFC encodes a fuller cognitive 'state' map (*Stalnaker et al., 2015*; *Wilson et al., 2014*; *Lopatina et al., 2017*).

Finally, it is worth noting that the current results are consistent with data showing that the OFC is necessary for performance in the final phase of training in this task, when information must be integrated to predict the reward. Neural activity in the probe test to the preconditioned cues clearly differed between pairs, and activity in the first cue of a pair appeared to encode the second cue, particularly for the critical AC cue pair. Activity to A was most similar to activity during the rewarded portions of B, and this coding was strongest in the rats that showed strong responding to A.

However, these data do not address whether the encoding of these associations in OFC during the preconditioning phase is necessary for performance in the final phase of training. The correlates in OFC may be merely a reflection of processing in other brain regions, such as the hippocampus and retrosplenial cortex, which are necessary in these earlier phases (*Robinson et al., 2014*). Consistent with this idea, the OFC receives strong input from hippocampus, which has a specific influence on the encoding in OFC in real time (*Wikenheiser et al., 2017*). In this case, temporary inactivation of OFC during the preconditioning phase should not affect inference in the final test. By contrast, representation of this information in OFC may be necessary in the preconditioning phase, perhaps to allow proper updating or integration with the new learning. If this is the case, then inactivation should affect later responding. Regardless, the identification of sensory-sensory representations in the OFC prior to their endowment with biological significance substantially expands the potential role of this area in this very simple and other more complex settings.

## Materials and methods

### Subjects

Twenty-one adult male Long-Evans rats (weighing 275–325 g on arrival) were individually housed and given ad libitum access to food and water, except during behavioral training and testing. During training and testing, they were restricted to 10 g of standard rat chow, which they received following each training session. Rats were maintained on a 12 hr light/dark cycle and trained and tested during the light cycle. Experiments were performed at the National Institute on Drug Abuse Intramural Research Program, in accordance with NIH guidelines. The number of subjects was chosen based on our expectations of what was needed to detect behavioral and neural evidence of learning on each experimental day (*Jones et al., 2012*).

### Apparatus

Behavioral training and testing were conducted in aluminum chambers, and cues and food reward were presented with commercially-available equipment (Coulbourn Instruments, Allentown, PA). A recessed food port was placed in the center of the right wall approximately 2 cm above the floor. The food port was attached to a pellet dispenser mounted outside the behavior chamber and delivered three small flavored sucrose pellets (Bioserve precision pellets) per rewarded cue presentation. Auditory cues (tone, siren, 2 Hz clicker, white noise) calibrated to ~65 dB were used during the behavioral testing.

### Surgical procedures

Rats underwent surgery for implantation of chronic recording electrode arrays. Rats were anesthetized with isoflurane and placed in a standard stereotaxic device. The scalp was excised, and holes were bored in the skull for the insertion of ground screws and electrodes. Multi-electrode bundles (16 nichrome microwires attached to a microdrive) were inserted 0.5 above orbitofrontal cortex [AP 3.2 mm and ML 3.0 mm relative to bregma (*Paxinos and Watson, 2009*); and DV 4.0 mm from the dura], unilaterally in 18 rats and bilaterally in two rats. One of the unilaterally implanted OFC rats had an additional electrode bundle implanted above the ipsilateral BLA (AP −3 mm, ML 5 mm relative to bregma; 7.0 mm from the dura). A reference wire for each bundle was wrapped around two skull screws in contact with dura. Once in place, the assemblies were cemented to the skull using dental acrylic, and electrodes were lowered into OFC over the course of surgical recovery. For 18 rats, behavioral training began 2–3 weeks following electrode implantation; an additional three subjects began training 10–14 weeks following electrode implantation, after participation in an olfactory operant task with liquid rewards.

### Behavioral training

The sensory preconditioning procedure consisted of three phases, of similar design to a prior study (*Jones et al., 2012*).

#### Preconditioning

Rats were shaped to retrieve pellets from a food port in one session; during this session, twenty pellets delivered over a 1 hr period. After this shaping, rats underwent 2 days of preconditioning. In each day of preconditioning, rats received trials in which two pairs of auditory cues (A→B and C→D) were presented in a blocked design. Each cue pair was presented six times. Cues were each 10 s long, the inter-trial intervals varied from 3 to 6 min, and the order the blocks was alternated across the two days. Cues A and C were a white noise or a clicker and cues B and D were a siren or a constant tone (counterbalanced). We experienced several equipment problems, which affected our data acquisition. Due to errors in a behavioral program, an excess trial for one or both cue pairs were presented in 14 of 42 sessions. These malfunctions were largely counterbalanced, with respect to which cue was over-presented, and findings from data in these sessions did not differ from the overall pattern of results. To incorporate these data into the main analysis, extra presentations on a given day for a given cue pair were excluded from neural and behavioral analysis. In addition, recording for one subject for the second preconditioning day was interrupted, forcing us to restrict the analysis to

the completed trials. Finally, behavior for one subject on the first preconditioning day was excluded because of data storage problems.

## Conditioning

After preconditioning, rats underwent conditioning. Each day, rats received a single training session, consisting of six trials of cue B paired with pellet delivery and six trials of D paired with no reward. The pellets were presented three times during cue B at 3, 6.5, and 9 s into the 10 s presentation of cue B. Cue D was presented for 10 s without reward. The two cues were presented in 3-trial blocks, counterbalanced. The inter-trial intervals varied between 3 and 6 min. The behavior for two subjects (one session from day three and one from day 6) was excluded because of data storage problems.

## Probe test

After conditioning, the rats underwent a single probe test, which consisted of three reminder trials of B paired with reward, interleaved with three trials of D unpaired. These were followed by blocked presentation of cues A and C, alone, six times each, without reward, and with the presentation of cue A or C first counterbalanced across subjects. Cue durations, timing of reward, and inter-trial intervals were as above.

## Electrophysiology

Neural signals were collected from the OFC during each behavioral session. Differential recordings were fed into a parallel processor capable of digitizing 16-to-32 signals at 40 kHz simultaneously (Plexon MAP). Discriminable action potentials of >3:1 signal/noise ratio were isolated on-line from each signal using an amplitude criterion in cooperation with a template algorithm. Discriminations were checked continuously throughout each session. Resultant timestamps and waveforms were saved digitally, and off-line re-analysis incorporating 3D cluster-cutting techniques were used to confirm and correct on-line discriminations.

## Statistical analyses

Data were processed with custom scripts and functions in Matlab R2014a, available online [*Sadacca, 2018*; copy archived at https://github.com/elifesciences-publications/OFC_SPC_17]. Conditioned responding was quantified by the percentage of time rats spent with their head in the food cup during cue presentation as measured by an infrared photo beam positioned at the front of the food cup. Magnitude of responding between pairs of cues was compared with a paired t-test. Spike times were sorted into bins and analyzed as specified. In comparing response differences evoked by different cues, bins spanning the full 10 s of cue-evoked activity were analyzed; in other analyses, smaller bins or sliding windows were utilized. In comparing fractions of neurons responding between conditions, a $2 \times 2$ chi-squared test for independence was used. In comparing relative neural responses, a Pearson linear correlation coefficient was calculated on this activity following a subtraction of average baseline activity (30 s before cue onsets), and correlation coefficients were compared following a Fisher r-to-z transformation. For probe-day neural data, analyses were restricted to the first two trials of A/C responding to capture the relationship among cue responses before behavioral extinction.

## Classification of neural data

For classifying individual preconditioning trials, a linear discriminant model was trained from a matrix of observations (all but one trial of each cue) and variables (a pseudo-ensemble of neurons of equivalent size to the number recorded that day, resampled with replacement from the population recorded on that day), using the average firing rate during a cue. This model was then tested on the held out trial and iterated 1000x. In addition to the classification of average activity, two control datasets were created to limit the influence of baseline difference in firing between AB trials and CD trials: one control used the average firing rate for a cue on a given trial minus the baseline on that trial, and a second control used the residual firing rates following a generalized linear regression of the average firing rates on the pre-cue baseline firing on that trial using a normal distribution. For classifying individual probe trials, a similar linear discriminant model was trained with a modification required by the reduced trial number. Here, we used a matrix of observations (all but one trial of

cues A and B) and variables (the first two principle components from a pseudo-ensemble of neurons of equivalent size to the number recorded that day, resampled with replacement from the population recorded on that day), using the average firing rate during cues A or C. Once trained on A/C trials, this model was tested on trials of cue B and D (projected into the PC space of the training data), scored for classification accuracy, and iterated 1000x.

## AUC normalization
In calculating AUC normalized firing rates for display purposes, we compared the histogram of spike counts during each bin of spiking activity (250 ms, test bins from each trial for a cue, at a particular time post-stimulus) against a histogram of baseline (250 ms) bins, from all trials for that cue. The ROC was calculated by normalizing all test and baseline bin counts, such that the minimum bin count was 0 and the maximal bin count was 1, and sliding a discrimination threshold across each histogram of bins, from 0 to 1 in. 01 steps, such that fraction of test bins identified above the threshold was a 'true positive' rate and the fraction of baseline bins above the threshold was a 'false negative' rate for an ROC curve. The area under this curve was then estimated by trapezoidal numerical estimation, with an auROC below. five being indicative of inhibition, and an auROC above. Five being indicative of excitation above baseline. For all statistical tests, an alpha level of 0.05 was used.

## Histology
After the final recording session, rats were euthanized and perfused first with PBS and then 4% formalin in PBS. Electrolytic lesions (1 mA for 10 s) made just before perfusion were examined in fixed, 0.05 mm coronal slices stained with cresyl violet. Anatomical localization for each recording session and final positioning was based on histology, stereotaxic coordinates of initial positioning, and recording notes.

## Acknowledgements
This work was supported by the Intramural Research Program at the National Institute on Drug Abuse. The opinions expressed in this article are the authors' own and do not reflect the view of the NIH/DHHS.

## Additional information

### Competing interests
Geoffrey Schoenbaum: Reviewing editor, *eLife*. The other authors declare that no competing interests exist.

### Funding

| Funder | Grant reference number | Author |
|---|---|---|
| National Institute on Drug Abuse | ZIA-DA000587 | Geoffrey Schoenbaum |

The funders had no role in study design, data collection and interpretation, or the decision to submit the work for publication.

### Author contributions
Brian F Sadacca, Conceptualization, Data curation, Formal analysis, Investigation, Writing—original draft, Writing—review and editing; Heather M Wied, Nina Lopatina, Conceptualization, Investigation, Methodology; Gurpreet K Saini, Daniel Nemirovsky, Investigation, Methodology; Geoffrey Schoenbaum, Conceptualization, Supervision, Funding acquisition, Writing—original draft, Project administration, Writing—review and editing

### Author ORCIDs
Geoffrey Schoenbaum  http://orcid.org/0000-0001-8180-0701

## Ethics

Animal experimentation: This study was performed in strict accordance with the recommendations in the Guide for the Care and Use of Laboratory Animals of the National Institutes of Health. All of the animals were handled according to approved institutional animal care and use committee (IACUC) protocols (#15-CNRB-108) of the NIDA-IRP. The protocol was approved by the Animal Care and Use Committee (Permit Number: A4149-01). All surgery was performed under gas anesthesia, and every effort was made to minimize suffering.

## Decision letter and Author response

Decision letter https://doi.org/10.7554/eLife.30373.011
Author response https://doi.org/10.7554/eLife.30373.012

# Additional files

## Supplementary files

• Transparent reporting form
DOI: https://doi.org/10.7554/eLife.30373.009

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
