## [Decision Letter]

Thank you for submitting your article "Orbitofrontal neurons signal sensory associations underlying model-based inference in a sensory preconditioning task" for consideration by *eLife*. Your article has been reviewed by two peer reviewers, and the evaluation has been overseen by Michael Frank as the Senior Editor and Reviewing Editor. The reviewers have opted to remain anonymous.

The reviewers have discussed the reviews with one another and the editor has drafted this decision to help you prepare a revised submission.

Summary:

The authors have shown in previous experiments that the orbitofrontal cortex (OFC) is critical for model-based behavior, including inference in the probe test of a sensory preconditioning (SPC) task. The current experiment addresses the question of whether OFC is necessary for inference because it encodes inferred values during the probe test, or whether the OFC plays a more general role in model- based behavior by encoding the associative structure of the task during preconditioning even when cues have not yet acquired value. Neurons in the rat OFC were recorded during all phases of a SPC task: preconditioning, conditioning, and probe test. The key question addressed here is whether OFC neurons only encode the value of the cues during conditioning and probe test, or whether they already encode the associative transition structure during preconditioning. The results clearly show that OFC neurons already encode associations between valueless cues during preconditioning. This is the key finding of this experiment and shows that the OFC supports model-based behavior by encoding the associative task structure. Additional results show that OFC neurons also encode the value of reward predictive cues during conditioning and that the same neurons that encode predicted value during the probe test also encode inferred value.

Essential revisions:

Reviewers agreed that this was an interesting paper with strong implications for our understanding of the OFC in model-based control of behavior, and with clever and sound experimental design. However, a number of issues were raised, especially pertaining to the analysis, which we would like you to address in a revision.

1) There was some discussion and mild disagreement amongst reviewers as to whether you should confine your analyses to data from animals that show behavioral sensory pre-conditioning effects. One reviewer thought that since the probe test is the only way to read out whether the animals have learned the association, the ones that do not aren't easily interpreted and should be confined to a supplement/comparison analysis at the end of the paper. The other reviewer felt that rather, learning cue-cue associations is a necessary but insufficient condition for responding to A>C in the probe test, and that you have increased power to detect effects by including all animals during preconditioning. But both reviewers agreed that you could address this issue by capitalizing on the variability across rats and assess whether you can predict, based on the sensory pre-conditioning or probe test neural activity, which rats will not show a behavioral effect, i.e. A will not predict B based on a classifier. Or, you could test whether there is a difference in the strength of encoding of cue-pairs during preconditioning in animals that later show inference compared to those that do not. Indeed you speculate that "the presence of these associations in OFC… suggests that they may be the substrate in OFC that is necessary for inference in the final probe test", so these analyses would allow you to support this assertion.

2) Compared to previous experiments by the same group (e.g., Jones et al.; Sadacca et al. 2016, *eLife*), responding to A vs. C during the probe test was relatively weak in the current experiment (and if anything might be indistinguishable from the OFC inactivated rats in the Jones study). Are there any differences in the experimental design that may explain this? If so, it would be informative to discuss this in the manuscript, as it would inform experimental conditions under which inference is enhanced/reduced in SPC tasks. This is somewhat disconcerting given the impressive numbers of neurons/animals recorded here. Compared to the usual number of animals in a neurophysiology report, the authors use a large number of animals (n=22). While the behavioral statistics take into account the variability between animals, the analysis of the neural data largely ignores this. I think that the authors should include animal as either a random or main effect in their analyses to account for inter-subject variability. This should be done irrespective of whether they decide to only analyze data from animals showing pre-conditioning effects.

Because of this I am wondering if the response to A is different to the response to D (i.e. another control stimulus that might be tangentially associated with reward through it being delivered in a rewarding context)? Other than a simple control analysis of A vs. D, this comment is motivated by the fact that neural responses to A and D are correlated, albeit weakly. In addition, statistical tests do not appear to be corrected for multiple comparisons (notably in Figure 6 where hundreds of classifications are performed and only a one-sided p-value of 0.05 is used to assess significance) and, related to my first issue, there is no assessment/test of whether effects were observed in the majority of animals recorded (e.g. Figure 2–Figure 5). More analyses are required to support the authors' statement and ensure that observed effects are reproducible.

3) It would be great to have recordings from sessions before the pre-conditioning phase to see that the responses to A and B are different in the first place and then converge during pre-conditioning. The decoding/classifier approach is good here, but is of course contaminated by the presence of B. One way to get at this could be to show pre-pre-conditioning responses if you have them or show raster plots from the very first preconditioning session where the responses should not overlap. Similarly, while it is convincing that these correlations increase from day 1 to day 2, suggesting that they are not merely driven by temporal proximity. However, I was wondering how much of the correlation is actually driven by temporal proximity. Would it be possible to estimate baseline correlations using two consecutive time windows from the inter-trial intervals?

4) "Although firing to B is contaminated by the delivery of reward at several points within the cue, the increased firing is evident in many neurons at the outset of cue B, and the firing does not seem to be specifically driven by reward delivery." I find it hard to verify this statement with the current analyses. On the plot, the change in neural activity is rarely at time 0 (although there are some like this), but later in time (+2/3 sec) I'm not sure it is appropriate to say so without isolating and quantifying responses to the cue only (from 0 to 2s) versus reward (2s/4.5s/7s).

5) Decoding Figure 3: "As illustrated in Figure 3 (top row, raw data), the population response showed a decline in self classification and an increase in within-pair classification across the two preconditioning days." This statement is ambiguous as it is not clear what is being decoded. Another reviewer agreed this is a little confusing but guessed that decoding is based on individual stimuli. Self-classification refers to "correct label" (e.g., A as A, B as B, C as C, and D as D) which is decreasing from day 1 to day 2 (black vs. gray line in Figure 3, right panel). At the same time, within-pair classification errors (A as B and C as D) go up (black vs. gray line in Figure 3, middle panel). But this needs to be clarified – e.g., there was confusion over the difference between self and pair decoding; please be up front about exactly what you did.

6) Figure 5: I am not comfortable with only the best and worst 5% being shown and analyzed in Figure 5. Surely it would be better to plot/analyze the whole population of neurons showing the effects as this is the principled approach. Using the whole population from only animals that show effects could be another approach. It is a little tricky to conclude an effect by only evaluating the extremes in a population. Without a proof that most of the A>C neurons responds more to B (than the C>A neurons), it is difficult to argue for a link between the neuronal activity and animals' behavior. The very small difference observed in Figure 5 suggests to me that there is no difference. This is also visible in Figure 5 where AUC differences for cue B exist nearly as much in these two populations of neurons (top vs. bottom) The authors do provide further decoding analyses on this point, but as for every decoding approach, only a handful of neurons could contribute to the decoding accuracy. This makes it vulnerable to the exact same issue as the previous 5% analysis. This needs to be tightened up.

7) Analyses of neuronal data focus on activity increases, rather than encoding (i.e., increases and decreases). For instance, neurons responding to cues during preconditioning are identified if they "significantly increased firing to at least one of the cues." It is unclear why the authors restricted their analysis to units that increased responding, as neurons might just as well encode cues by significantly decreasing firing in response to the cue. The same applies to the analysis of data from conditioning, which focuses on units in which activity to the reward predictive cue increased significantly over the course of 6 days of conditioning.

---

## [Author Response]

Essential revisions:Reviewers agreed that this was an interesting paper with strong implications for our understanding of the OFC in model-based control of behavior, and with clever and sound experimental design. However, a number of issues were raised, especially pertaining to the analysis, which we would like you to address in a revision.1) There was some discussion and mild disagreement amongst reviewers as to whether you should confine your analyses to data from animals that show behavioral sensory pre-conditioning effects. One reviewer thought that since the probe test is the only way to read out whether the animals have learned the association, the ones that do not aren't easily interpreted and should be confined to a supplement/comparison analysis at the end of the paper. The other reviewer felt that rather, learning cue-cue associations is a necessary but insufficient condition for responding to A>C in the probe test, and that you have increased power to detect effects by including all animals during preconditioning. But both reviewers agreed that you could address this issue by capitalizing on the variability across rats and assess whether you can predict, based on the sensory pre-conditioning or probe test neural activity, which rats will not show a behavioral effect, i.e. A will not predict B based on a classifier. Or, you could test whether there is a difference in the strength of encoding of cue-pairs during preconditioning in animals that later show inference compared to those that do not. Indeed you speculate that "the presence of these associations in OFC… suggests that they may be the substrate in OFC that is necessary for inference in the final probe test", so these analyses would allow you to support this assertion.

We appreciate the interest in this question. Generally, our feeling is that it would be an open question whether there was any relationship between the development of S-S coding in the preconditioning phase in the OFC and subsequent evidence of this learning in food-cup directed behavior in the probe test. To put this another way, our basic result – that there is acquisition of the S-S learning in OFC – could be directly related to later food cup behavior (necessary) or it could be completely unrelated (necessary but insufficient); or something in the middle. Figuring this out is complicated by the fact that we have no evidence of learning in the preconditioning phase other than the food cup responding, and obviously, this is just one measure. There are likely many reasons we would not see A>C behavior in the probe test by this simple metric, many of which may be more likely than the idea that the rat literally failed to notice that A and B were paired. This actually seems extremely unlikely to us.

Unfortunately, even with 22 subjects, we do not have enough data to say for sure how preconditioning encoding is related to probe test behavior. While all analyses produce nearly identical results if we exclude rats that do not show evidence of learning in the probe test (see supplement for the preconditioning Figure 2 and Figure 3), the same pattern is also generally present in the data from the poor performers. It is perhaps weaker, but it is hard to tell, as we do not have enough statistical power in the remaining data to make comparisons. In addition, regressions against final behavior weren’t informative.

So while we agree that this is an interesting question, our data do not provide a definitive answer. Rather than providing one that is confusing, we have instead highlighted this as a question in our discussion. This preserves the main point of the study – that OFC neurons acquire these representations, which they should not if OFC neurons only represent value or biologically meaningful associative information – while raising this as a future question.

2) Compared to previous experiments by the same group (e.g., Jones et al.; Sadacca et al. 2016, eLife), responding to A vs. C during the probe test was relatively weak in the current experiment (and if anything might be indistinguishable from the OFC inactivated rats in the Jones study). Are there any differences in the experimental design that may explain this? If so, it would be informative to discuss this in the manuscript, as it would inform experimental conditions under which inference is enhanced/reduced in SPC tasks. This is somewhat disconcerting given the impressive numbers of neurons/animals recorded here. Compared to the usual number of animals in a neurophysiology report, the authors use a large number of animals (n=22). While the behavioral statistics take into account the variability between animals, the analysis of the neural data largely ignores this. I think that the authors should include animal as either a random or main effect in their analyses to account for inter-subject variability. This should be done irrespective of whether they decide to only analyze data from animals showing pre-conditioning effects.

We appreciate this concern. However, while overall responding to cues was less (for the reasons outlined below), there was still significantly more responding to the critical cues (A and B) than to the control cues (C and D) – and in a ratio similar to previous studies. The effect size between Jones 2012 and this study likely differs because of the neural recordings: it’s our general observation that the cables connecting subjects for neural data acquisition modestly decrease overall responding to conditioned cues. In addition, this study differs from Sadacca 2016 in the method of reinforcement: this study uses solid food pellets; Sadacca 2016 used liquid reward. While normal between-group variability can account for the difference between overall levels of responding, subtle differences in food vs. water restriction likely increased normal between group variability. To give a better view to exactly how individual behavioral responses differed among subjects, we’ve changed a binned histogram of individual responses of A-C to a scatter of responding to A vs. C.

As for the number of subjects, this was driven by what was necessary to show the behavioral effect as well as what was required to collect data from a sufficient number of neurons. For behavior, a minimum of 15 subjects is required (e.g. Jones, Sadacca Sharpe et al. – 16, 14, 18/19 per group, respectively). For recordings, subjects were only run once through the experimental protocol, and with an average of ~5 neurons per subject per day, 22 subjects were required to be confident in acquiring >100 neurons each recording day.

Because of this I am wondering if the response to A is different to the response to D (i.e. another control stimulus that might be tangentially associated with reward through it being delivered in a rewarding context)? Other than a simple control analysis of A vs. D, this comment is motivated by the fact that neural responses to A and D are correlated, albeit weakly.

We appreciate the reviewers’ interest in A/D, but we don’t believe these cues are comparable for obvious reasons. Instead the appropriate comparison for preconditioning, going back to the original studies by Brogden, is between A and C, which have been treated identically. Indeed while there is a modest A/D correlation, which presumably reflects general auditory responsiveness or generalization since all the cues are intentionally very similar, the correlation between A and B is much higher, grows with learning, and predicts behavior.

In addition, statistical tests do not appear to be corrected for multiple comparisons (notably in Figure 6 where hundreds of classifications are performed and only a one-sided p-value of 0.05 is used to assess significance).

While individual time points for this analysis are the subjects of several tests (with the 100’s of simulations are observations of the underlying distribution at each time), the comparisons here might be more conservative than then reviewers suspect. The likelihood of adjacent data points being well-classified by chance is p>0.0025 – and the likelihood of 5 seconds of bins during the reward period being significant as observed is miniscule. In contrast, if this bin-by-bin permutation test was permissive, we would expect several individual bins aside from those ‘reward’ bins to be significant, whereas they are neither elsewhere in animals with good nor poor behavior, save a single bin.

And, related to my first issue, there is no assessment/test of whether effects were observed in the majority of animals recorded (e.g. Figure 2–Figure 5). More analyses are required to support the authors' statement and ensure that observed effects are reproducible.

We agree the observed results are stronger with tests across individual ensembles instead of pseudo-ensemble. For the preconditioning data, we have the power to resolve such relationships: in simulating pseudo ensembles, ensembles of ~10 neurons were required, and an ANOVA across correlations shows a significant effect of cue-pair across pairs AB, AD, CB, and CD (F = 3.88, p = 0.012), and a t-test across ensembles comparing mean correlation within pair (AB/CD) vs. between pair (AD/CB) showed paired correlations significantly greater than unpaired correlations (t = 2.82 p= 0.01). For the probe data (Figure 5), however, we do not have the ensemble sizes required to resolve these effects: in simulating pseudoensembles, ensembles of >50 neurons were required to reliably resolve a greater correlation between paired than unpaired cues and no ensembles from the probe session exceeded 25 neurons.

3) It would be great to have recordings from sessions before the pre-conditioning phase to see that the responses to A and B are different in the first place and then converge during pre-conditioning. The decoding/classifier approach is good here, but is of course contaminated by the presence of B. One way to get at this could be to show pre-pre-conditioning responses if you have them or show raster plots from the very first preconditioning session where the responses should not overlap. Similarly, while it is convincing that these correlations increase from day 1 to day 2, suggesting that they are not merely driven by temporal proximity. However, I was wondering how much of the correlation is actually driven by temporal proximity. Would it be possible to estimate baseline correlations using two consecutive time windows from the inter-trial intervals?

We agree that mere contiguity might be an explanation for the correlated firing. The reviewers’ suggestion to pre-expose the rats to the cues and record prior to the preconditioning is obviously excellent from a neurophysiology point of view. However behaviorally it is likely to have unpredictable if not disastrous effects, since essentially that turns our experiment into a test of latent inhibition of S-S learning. We might see much weaker or no learning at all. For that reason, we chose not to record during pre-exposure so we cannot provide the data requested. We believe that the fact that the correlated activity increases across days and for the appropriate cue pairings shows that at least some of the correlation is associative in nature.

In addition we now provide tests of correlations between adjacent bins of cue responses (i.e. first half of first cue vs. last half of first cue to bins of cue 2) in Figure 2—figure supplement 2 and in the main text (subsection “Orbitofrontal neurons acquire ability to distinguish cue pairs during preconditioning”, second paragraph) as a comparison of the strength of adjacency on the observed correlations, similar to the suggestion made by the reviewer above. These data showed no significant difference between early/late in cues and the subsequent cue, bolstering our contention that the correlated firing is not simply contiguity based.

4) "Although firing to B is contaminated by the delivery of reward at several points within the cue, the increased firing is evident in many neurons at the outset of cue B, and the firing does not seem to be specifically driven by reward delivery." I find it hard to verify this statement with the current analyses. On the plot, the change in neural activity is rarely at time 0 (although there are some like this), but later in time (+2/3 sec) I'm not sure it is appropriate to say so without isolating and quantifying responses to the cue only (from 0 to 2s) versus reward (2s/4.5s/7s).

We now explicitly do this analysis, and report these numbers in the revised Results subsection “Orbitofrontal neurons acquire the ability to predict reward during Pavlovian conditioning”.

5) Decoding Figure 3: "As illustrated in Figure 3 (top row, raw data), the population response showed a decline in self classification and an increase in within-pair classification across the two preconditioning days." This statement is ambiguous as it is not clear what is being decoded. Another reviewer agreed this is a little confusing but guessed that decoding is based on individual stimuli. Self-classification refers to "correct label" (e.g., A as A, B as B, C as C, and D as D) which is decreasing from day 1 to day 2 (black vs. gray line in Figure 3, right panel). At the same time, within-pair classification errors (A as B and C as D) go up (black vs. gray line in Figure 3, middle panel). But this needs to be clarified – e.g., there was confusion over the difference between self and pair decoding; please be up front about exactly what you did.

We apologize for the lack of clarity in this figure, though the reviewer's guess was exactly right. We’ve extended our description of this analysis to improve clarity in the fifth paragraph of the subsection “Orbitofrontal neurons acquire ability to distinguish cue pairs during preconditioning”.

6) Figure 5: I am not comfortable with only the best and worst 5% being shown and analyzed in Figure 5. Surely it would be better to plot/analyze the whole population of neurons showing the effects as this is the principled approach. Using the whole population from only animals that show effects could be another approach. It is a little tricky to conclude an effect by only evaluating the extremes in a population. Without a proof that most of the A>C neurons responds more to B (than the C>A neurons), it is difficult to argue for a link between the neuronal activity and animals' behavior. The very small difference observed in Figure 5 suggests to me that there is no difference. This is also visible in Figure 5 where AUC differences for cue B exist nearly as much in these two populations of neurons (top vs. bottom) The authors do provide further decoding analyses on this point, but as for every decoding approach, only a handful of neurons could contribute to the decoding accuracy. This makes it vulnerable to the exact same issue as the previous 5% analysis. This needs to be tightened up.

While we only displayed the mean activity of the 5% of the neurons best discriminating cues A from C, it in fact doesn't matter what fraction we show, and have instead plotted a larger% . We stress, though, that this plot was intended to illustrate a general feature of the data and wasn’t intended as a fundamental analysis. We would also like to stress that for decoding approaches, we resample activity with small ensembles to test robustness to just a handful of neurons underlying the effect, but include all neurons in the decoding analysis.

7) Analyses of neuronal data focus on activity increases, rather than encoding (i.e., increases and decreases). For instance, neurons responding to cues during preconditioning are identified if they "significantly increased firing to at least one of the cues." It is unclear why the authors restricted their analysis to units that increased responding, as neurons might just as well encode cues by significantly decreasing firing in response to the cue. The same applies to the analysis of data from conditioning, which focuses on units in which activity to the reward predictive cue increased significantly over the course of 6 days of conditioning.

In much of the manuscript, we show data without regard to whether neurons increased or suppressed firing in response to the cues. For example, in the scatter plots, we illustrate changes for both increasing and suppressed populations. Consistent with what we have always found previously, the effects evident in the two populations are basically identical or mirror images (e.g. Ogawa et al., 2013). However there are generally not enough cells that suppress firing to conduct a formal analysis comparing the two groups. This is because of the lower parametric space in which to see decreases in firing. This also affects the ability of these neurons to show differential firing, although there is some indication from our other studies that the prevalence of neurons that suppress firing does not increase with conditioning, suggesting these neurons may not be representing associative information the same way as excitatory neurons (Takahashi et al., 2013). However in none of our analyses of neural responses are these neurons excluded. If the reviewers have a specific issue or question where they think it is important, we would be happy to do any particular analysis. But we have not included parallel analyses for neural subtypes everywhere that they could be done, since they were not informative in our opinion and would serve only to make the paper less comprehensible.